# Role of Signal Transduction Pathways and Transcription Factors in Cartilage and Joint Diseases

**DOI:** 10.3390/ijms21041340

**Published:** 2020-02-17

**Authors:** Riko Nishimura, Kenji Hata, Yoshifumi Takahata, Tomohiko Murakami, Eriko Nakamura, Maki Ohkawa, Lerdluck Ruengsinpinya

**Affiliations:** Department of Molecular & Cellular Biochemistry, Osaka University Graduate School of Dentistry1-8 Yamadaoka, Suita, Osaka 565-0871, Japan; hata@dent.osaka-u.ac.jp (K.H.); takahata@dent.osaka-u.ac.jp (Y.T.); tmurakami@dent.osaka-u.ac.jp (T.M.); enakamura@dent.osaka-u.ac.jp (E.N.); m-ohkawa@dent.osaka-u.ac.jp (M.O.); rlerdluck@dent.osaka-u.ac.jp (L.R.)

**Keywords:** osteoarthritis, rheumatoid arthritis, fibrodysplasia ossificans progressive, achondroplasia

## Abstract

Osteoarthritis and rheumatoid arthritis are common cartilage and joint diseases that globally affect more than 200 million and 20 million people, respectively. Several transcription factors have been implicated in the onset and progression of osteoarthritis, including Runx2, C/EBPβ, HIF2α, Sox4, and Sox11. Interleukin-1 β (IL-1β) leads to osteoarthritis through NF-ĸB, IκBζ, and the Zn^2+^-ZIP8-MTF1 axis. IL-1, IL-6, and tumor necrosis factor α (TNFα) play a major pathological role in rheumatoid arthritis through NF-ĸB and JAK/STAT pathways. Indeed, inhibitory reagents for IL-1, IL-6, and TNFα provide clinical benefits for rheumatoid arthritis patients. Several growth factors, such as bone morphogenetic protein (BMP), fibroblast growth factor (FGF), parathyroid hormone-related protein (PTHrP), and Indian hedgehog, play roles in regulating chondrocyte proliferation and differentiation. Disruption and excess of these signaling pathways cause genetic disorders in cartilage and skeletal tissues. Fibrodysplasia ossificans progressive, an autosomal genetic disorder characterized by ectopic ossification, is induced by mutant ACVR1. Mechanistic target of rapamycin kinase (mTOR) inhibitors can prevent ectopic ossification induced by ACVR1 mutations. C-type natriuretic peptide is currently the most promising therapy for achondroplasia and related autosomal genetic diseases that manifest severe dwarfism. In these ways, investigation of cartilage and chondrocyte diseases at molecular and cellular levels has enlightened the development of effective therapies. Thus, identification of signaling pathways and transcription factors implicated in these diseases is important.

## 1. Introduction

Cartilage is an important tissue that functions in skeletal development, tissue patterning, locomotive organs, respiration, and organ support [1,2]. Two very common diseases of cartilage and joints, osteoarthritis and rheumatoid arthritis, are mainly caused by excessive mechanical stress, inflammation, and immunological changes [3,4]. As biochemical and genetic studies have dramatically progressed in the past decade, the molecular basis of osteoarthritis and rheumatoid arthritis has become clearer. Signaling molecules and transcription factors involved in the pathogenesis of osteoarthritis have been identified [5], although the development of effective reagents for osteoarthritis remains complicated. Interleukin-1 (IL-1) has been implicated in the development of osteoarthritis through its intracellular signaling pathways and downstream transcriptional events [6]. Several transcription factors, including Runt-related transcription factor 2 (Runx2) and the group C of sex determining region Y-box (SoxC) family members, have been demonstrated to regulate expression of the degradative enzymes of cartilage matrices, which results in the onset of osteoarthritis [7,8]. Moreover, micro-RNA (miR) are reportedly associated with the pathogenesis of osteoarthritis by modulating transcriptional machinery [9,10]. Additionally, recent studies have indicated a role for Wnt family members in the pathogenesis of osteoarthritis [11]. 

Based on accumulating knowledge that some inflammatory cytokines, such as tumor necrosis factor α (TNFα), IL-1, and IL-6, play critical roles in the onset and progression of rheumatoid arthritis, specific inhibitory reagents for these cytokines have been developed [12,13]. For instance, anti-TNFα antibodies, soluble TNF receptor 2-conjugated Fc domain of IgG1, anti-IL-6 receptor antibody, and an IL-1 receptor antagonist have exhibited dramatic effects on rheumatoid arthritis patients [13,14]. Furthermore, specific inhibitors of Janus kinases (JAKs), the downstream tyrosine kinases of IL-6, are also effective for rheumatoid arthritis patients [15,16]. Likewise, our understanding of the molecular basis of osteoarthritis and rheumatoid arthritis has progressed, and this contributes to better understanding of the pathogenesis of these diseases.

Several growth factors and cytokines, including bone morphogenetic protein (BMP), Indian hedgehog (Ihh), fibroblast growth factor (FGF), and parathyroid hormone-related protein (PTHrP), precisely regulate chondrocyte proliferation and differentiation [2,17]. However, disruption and excess of these signaling pathways can cause genetic disorders in cartilage and skeletal tissues. For example, mutations in *ACVR1*, *FGFR3*, *PTHrP*, and *GLI2* genes result in fibrodysplasia ossificans progressive (FOP) [18,19], achondroplasia (ACH) [20], brachydactyly type E2 [21], and Culler–Jones syndrome [22], respectively. Recent investigations have revealed the molecular basis of FOP and ACH, thus providing potential therapeutic approaches.

Understanding the molecular basis of cartilage and joint diseases is important and useful for the establishment of effective therapies. Thus, in this review, we introduce and discuss our recent understanding of the role of intracellular signaling pathways and transcription factors involved in cartilage and joint diseases, as well as potential therapies.

## 2. Role of Interleukin-1 (IL-1) and Nuclear Factor ĸB (NF-ĸB) Signaling in Osteoarthritis

Similar to rheumatoid arthritis, inflammatory cytokines, such as IL-1 and TNFα, are likely deeply implicated in the onset of osteoarthritis, although, it remains controversial whether osteoarthritis is an inflammatory disease. NF-ĸB functions as a major downstream signaling effector of IL-1. A component of NF-ĸB, RelA (also known as p65 proto-oncogene NF-ĸB subunit) reportedly regulates expression of a disintegrin and metalloproteinase with thrombospondin motifs 5 (Adamts5) in chondrocytes. Moreover, p65/RelA has been implicated in osteoarthritis development [23]. In murine experimental models, NF-ĸB activity was correlated with the degree of pain induced by osteoarthritis [24,25]. Moreover, a clinical study observed increased RELA expression in articular chondrocytes of patients with osteoarthritis [26]. NF-ĸB signaling also plays a major role in IL-1-dependent osteoarthritis [27]. Recently, expression of Yes-associated protein (YAP), a major mediator of Hippo signaling, was found to be decreased along with the development of osteoarthritis, whereas YAP activation inhibited the degradation of cartilage tissues in an osteoarthritis model [28]. Additionally, YAP deficiency enhanced expression of *Adamts4*, *Adamts5*, matrix metallopeptidase 3 (*MMP3*), and *MMP13*, and promoted cartilage destruction [28]. Moreover, YAP suppressed NF-κB signaling through transforming growth factor-β (TGF-β)-activated kinase 1 and NF-κB inhibitor alpha kinase [28]. Therefore, YAP has opposing effects on NF-ĸB signaling in the pathogenesis of osteoarthritis. 

Nuclear factor of κ light polypeptide gene enhancer in B cells inhibitor zeta (IκBζ) was initially proposed to function as an inhibitory molecule for NF-κB; however, it has been shown that IκBζ also mediates inflammatory signals downstream of IL-1 and TNFα [29,30]. IL-1β upregulated IκBζ expression in chondrocytes, and IκBζ expression was found to be increased in cartilage tissues of patients with osteoarthritis [31]. Cartilage-specific IκBζ-conditional knockout (KO) mice exhibited suppressed development of osteoarthritis. Moreover, IκBζ was shown to form a complex with NF-ĸB [32]. Thus, IκBζ appears to be a novel player in the pathogenesis of inflammatory stimuli-induced osteoarthritis [31]. 

IL-1β increased Zn^2+^ influx and markedly induced expression of the Zn^2+^ importer bZIP transcription factor 8 (ZIP8) in chondrocytes [33]. Expression of ZIP8 was also found to be upregulated in cartilage of patients with osteoarthritis [33]. Knockdown of ZIP8 suppressed IL-1β-induced expressions of *MMP3* and *MMP13* [33]. In mice, ZIP8 overexpression increased *MMP13* expression and osteoarthritis phenotypes, whereas ZIP8 deficiency markedly reduced the development of osteoarthritis. Most importantly, metal regulatory transcription factor 1 (MTF1) was identified as a transcriptional regulator of ZIP8, and osteoarthritis phenotypes were attenuated by MTF1 deficiency in cartilage [33]. Together, these results demonstrate that the linkage between IL-1 and the Zn^2+^-ZIP8-MTF1 axis is important in the onset of osteoarthritis. 

## 3. Role of Runx2 and Related Transcription Factors in Osteoarthritis

Hypertrophy of chondrocytes is often observed in patients with osteoarthritis [34]. Moreover, osteophyte formation and cartilage destruction observed in these patients resemble the calcification and degradation of cartilage matrices during late-stage endochondral ossification [34]. Thus, several investigations have focused on molecules involved in endochondral ossification. Runx2 plays a critical role in chondrocyte hypertrophy, as dissection of Runx2-KO mice revealed very few hypertrophic chondrocytes [35]. The functional role of Runx2 in cartilage development is partly compensated by Runx3 [35]. Accordingly, hypertrophy of chondrocytes was not observed in mice deficient for both Runx2 and Runx3. Consistent with these findings, Runx2 has been implicated in osteoarthritis following the observation that heterozygous Runx2-deficient mice are more resistant to osteoarthritis treatment than wild-type mice [36]. Recently, the progression of osteoarthritis was found to be accelerated in chondrocyte-specific Runx2-overexpressing mice [37]. Furthermore, methylation level of the *RUNX2* gene has been correlated with the risk for osteoarthritis [38]. However, there is some debate regarding the pathogenic effects of RUNX2 in osteoarthritis because no reported evidence suggests the altered occurrence of osteoarthritis in patients with cleidocranial dysplasia compared with other populations (personal communication). One explanation for this discrepancy might be that different species have different *Runx2* expression patterns. In contrast to Runx2, Runx1 has been shown to exhibit opposite functions in osteoarthritis. Indeed, osteoarthritis phenotypes, including osteophyte formation and cartilage destruction, were reduced by injection of *Runx1* mRNA in mice [39]. Similarly, a compound was identified to inhibit osteoarthritis through upregulation of *Runx1* [40]. Although the molecular basis is currently unclear, Runx1 is a potential therapeutic target for osteoarthritis.

As the transcriptional partner of Runx2 during cartilage development, C/EBPβ is thought to also have a role in the pathogenesis of osteoarthritis. C/EBPβ has been demonstrated to physically and functionally interact with Runx2 [41]. Additionally, C/EBPβ can reportedly induce expression of *MMP13*, a major collagenase for type 2 collagen, in cooperation with Runx2, consequently promoting osteoarthritis [42]. A biochemical study indicated that C/EBPβ binds to the *MMP13* gene to induce its expression in articular chondrocytes [42]. Moreover, C/EBPβ seems to play a role in regulation of MMP3 and Adamts5 proteins, both of which function as major degradative enzymes of cartilage matrices [43]. A human study demonstrated the relationship between osteoarthritis and polymorphisms of the *C/EBPβ* gene [42]. Thus, C/EBPβ appears to be implicated in the onset of osteoarthritis in cooperation with Runx2.

Hypoxia-inducible factor 2 α (HIF2α), an upstream target of C/EBPβ, can reportedly degrade articular cartilage [42]. In addition, roles of HIF2α in regulation of *MMP13* and *VEGF* expression have been reported [44]. Heterozygous HIF2α-deficient mice are resistant to osteoarthritis treatment [44,45], and exhibit reduced hypertrophic chondrocytes and *MMP13* expression [44]. Moreover, a single nucleotide polymorphism (SNP) of the *HIF2α* gene has been implicated in the development of osteoarthritis [45]. Interestingly, HIF2α has been shown to exhibit reciprocal activation with the Zn^2+^-ZIP8-MTF1 axis [46]. However, in contrast to these reports, two studies found no relationship between HIF2α and osteoarthritis. A human genomic study described no correlation between osteoarthritis and SNPs of the *HIF2α* gene [47]. Another study disputed the pathogenic role of HIF2α in osteoarthritis by showing that chondrocyte-specific HIF2α conditional-KO mice exhibited only marginally impaired endochondral ossification [48]. Thus, the involvement of HIF2α in osteoarthritis remains controversial.

As described above, several transcription factors regulate expression of *MMP13*, which plays an important role in endochondral ossification and the onset of osteoarthritis. Osterix has been shown to associate with Runx2 and directly regulate *MMP13* expression [49]. Indeed, *MMP13* expression is abolished in both global and cartilage-specific conditional Osterix-KO mice [49]. Moreover, Osterix-KO mice exhibited no ossification of cartilage matrices and lacked matrix vesicle formation [49,50,51]. Although the involvement of Osterix in osteoarthritis has not yet been investigated, it has been implicated in the pathogenesis of osteoarthritis [51]. 

## 4. Micro-RNA 140 (miR-140) and Osteoarthritis

miR play important roles in physiological and pathological events. Expression of miR-140, which is located in the intron of the WW domain-containing E3 ubiquitin protein ligase 2 (*Wwp2*) gene, increases with chondrocyte differentiation [52,53]. Originally identified as a highly expressed gene in cartilage, *Wwp2* is involved in cartilage development [53] and reduced in osteoarthritic cartilage compared with normal cartilage [10]. Mechanistically, IL-1 treatment has been shown to downregulate *Wwp2* expression in articular chondrocytes [52]. These findings indicate an anti-pathological role of Wwp2 and/or miR-140 in osteoarthritis. Consistent with these findings, miR-140-deficient mice, in which the host *Wwp2* gene was intact, exhibit abnormal skeletal formation and age-related osteoarthritis phenotypes [10]. As *Adamts5* expression was increased in these mice [10], it was suggested that Adamts5 is a direct target of miR-140. Indeed, in vitro analyses indicated that miR-140 targeted the 3ʹ untranslated region of the *Adamts5* gene [10]. Moreover, transgenic mice overexpressing miR-140 in chondrocytes were shown to be resistant to treatment for antigen-induced arthritis. A more precise analysis of Wwp2 and miR-140 recently reported that Wwp2 mutant mice had no craniofacial abnormalities [53], in contrast to a previous study that observed impaired skull formation in Wwp2-KO mice [54]. Together, these results indicate that miR-140, rather than Wwp2, is crucial for craniofacial development and chondrogenesis in addition to its suggested role in cartilage homeostasis [55]. These investigations highlight the difficulties in dissecting the functions of host genes and their miR.

Interestingly, nuclear factor of activated T cells (NFAT) 3, an NFAT family transcription factor, and TGF-β signaling have been implicated in regulation of miR-140 but not *Wwp2* in patients with osteoarthritis [56]. Another NFAT transcription factor, NFAT1, was shown to elicit protective effects against osteoarthritis, as NFAT1-KO mice showed sensitivity to osteoarthritis [57,58]. Mechanistically, histone methylation of the *NFAT1* gene at H3K9 is a key regulator of NFAT1 expression in chondrocytes [57]. Deficiency of NFAT1 and NFAT2 caused early osteoarthritis in mice [59], suggesting that NFAT family members are important for the maintenance of cartilage and joints. TGF-β/Smad signaling has also been associated with osteoarthritis. TGF-β signaling was stimulated in the subchondral bones of an osteoarthritis experimental model, whereas blockade of TGF-β signaling effectively inhibited osteoarthritis development [60]. Although the whole picture of miR-140, NFAT3, and TGF-β in osteoarthritis sounds elusive and complex, defining this network would contribute to the development of therapeutic targets for osteoarthritis.

## 5. Involvement of Wnt Signaling in Osteoarthritis

Canonical Wnt signaling plays a role in endochondral ossification. β-catenin, a central signaling mediator of canonical Wnt, interacts with sex determining region Y-box 9 (Sox9), an essential transcription factor for chondrogenesis, to promote its degradation through the proteasome-ubiquitin system and regulate endochondral ossification [61]. In addition to cartilage development, canonical Wnt signaling has been implicated in the pathogenesis of osteoarthritis. Transgenic mice overexpressing β-catenin in cartilage manifested an osteoarthritis-like phenotype [62]. Moreover, a genetic linkage has been shown between osteoarthritis and a SNP of the frizzled related protein (*FRZB*) gene, an antagonist of canonical Wnt signaling [63]. Consistently, *FrzB*-deficient mice showed resistance to osteoarthritic induction compared with wild-type mice [64]. Similarly, loss of sclerostin, a secreted inhibitory protein for canonical Wnt signaling, can reportedly promote osteoarthritis [65]. More recently, sclerostin was shown to inhibit expression of *MMP2* and *MMP3*, and protect against posttraumatic osteoarthritis [66]. Dickkpf-1 (DKK1), another antagonist of canonical Wnt signaling, is involved in remodeling of joints and bone in association with TNFα [67]. Treatment with an anti-DKK1 antibody suppressed bone erosion and increased osteophyte formation in an osteoarthritis model. In contrast, canonical Wnt signaling was found to be required for maintenance of the biological features and proliferative activity of articular chondrocytes [68]. Collectively, these studies indicate that the degree of canonical Wnt signaling must be appropriately regulated to maintain the physiological condition of cartilage. 

Non-canonical Wnt ligands, such as Wnt5a, appear to aggravate osteoarthritis. Degradation of type 2 collagen in chondrocytes was reduced by knockdown of Wnt5a [69]. In addition, Wnt5a was found to upregulate MMP expression in articular chondrocytes through catabolic signaling [70]. Therefore, non-canonical Wnt signaling may be a risk factor for osteoarthritis.

## 6. Osteoarthritis and SoxC Transcription Factor

In recent years, the unique features of articular chondrocytes compared with growth plate chondrocytes have begun to be elucidated. Moreover, the transcription factors involved in degradation of articular chondrocytes have been investigated. Although the SoxC family of transcription factors are important for endochondral ossification [71], *Sox4* and *Sox11*, were shown to be specifically upregulated by retinoic acid and IL-1 treatment in articular chondrocytes [8]. Sox4 and Sox11 upregulate expression of *Adamts4* and *Adamts5* by directly binding to their promoters [8]. Overexpression of Sox4 and Sox11 leads to the degradation of articular cartilage and increased expression of *Adamts5* and *MMP13* [8]. Furthermore, expression of *SOX4* and *SOX11* was associated with the destruction of cartilage in patients with osteoarthritis [8], suggesting the importance of these transcription factors in both the onset and progression of osteoarthritis. However, Sox11 could regulate growth differentiation factor 5 expression in chondrogenic cell lines ATDC5 and C3H10T1/2 [72]. Considering the important role of SoxC family members in endochondral ossification [71], Sox4 and Sox11 are likely to possess two different functions in physiological and pathological conditions. 

## 7. Inflammatory Cytokine Signaling and Rheumatoid Arthritis

Rheumatoid arthritis is a common immunological and inflammatory disease of cartilage and joints. Several inflammatory cytokines, such as TNFα, IL-1, IL-6, IL-17, and receptor activator of nuclear factor-kappa B ligand (RANKL), play an important role in the pathogenesis of rheumatoid arthritis [73,74]. Although rheumatoid arthritis is considered to primarily result from a pathological imbalance of immune cells and the aforementioned inflammatory cytokines, features of chondrocytes themselves are also implicated in disease onset and progression. As TNFα, IL-1, and IL-6 affect chondrocytes during rheumatoid arthritis development, their intracellular signaling pathways, including NF-ĸB and JAK/signal transducer and activator of transcription (STAT) signaling cascades, play a central role in destruction of cartilage tissues (Figure 1). IL-1 induced *MMP1* and *MMP13* expression in chondrocytes through p38, c-Jun NH2-terminal kinase (JNK), and NF-ĸB [75]. Similarly, in articular chondrocytes, TNFα upregulated *MMP13* via extracellular regulated mitogen-activated protein kinase (ERK), p38, JNK, NF-ĸB, and activating protein-1 (AP-1) [76]. Moreover, NF-ĸB and AP-1 signals activated by IL-17 reportedly cause damage to chondrocytes by regulating *MMP3*, *MMP13*, and *Adamts4* expression [77]. 

JAK/STAT and mitogen-activated protein kinase (MAPK) pathways activated by IL-6 have been shown to increase *MMP1*, *MMP3*, and *MMP13* in human chondrocytes [78]. Interestingly, oncostatin M, an IL-6 superfamily member, was upregulated in patients with rheumatoid arthritis, whereby it was found to increase *MMP1*, *MMP3*, and *MMP13* through ERK1/2, p38, JNK, and JAK/STAT pathways in chondrocytes [79]. Blockade of oncostatin M suppressed expression of these MMPs and related changes in signaling pathways [79]. JAKs have been identified as a therapeutic target for rheumatoid arthritis [15,80]. Tofacitinib and baricitinib, two specific JAK inhibitors that have been used as therapeutics for rheumatoid arthritis [15,80], seemingly exert their pharmacological effects by blocking IL-6/JAK/STAT signaling (Figure 1). Notably, a recent study used CRISPR/Cas9 technology to reveal that STAT3 binds a potential enhancer far upstream of the *Sox9* gene and is involved in chondrogenesis [81]. However, physiological and pathological roles of STAT3 in chondrocytes need further investigation.

RANKL has also been implicated in the onset and progression of rheumatoid arthritis by activating osteoclast development and function [74]. However, RANKL seems to have no effect on cartilage, presumably because RANK (the receptor of RANKL) is not expressed in articular chondrocytes. Indeed, the anti-RANKL antibody, denosumab, inhibits bone resorption, but not cartilage destruction, in patients with rheumatoid arthritis [82]. Thus, blockade of the RANKL signal is not sufficient to cure rheumatoid arthritis.

## 8. Transcription Factors Involved in Rheumatoid Arthritis

Most investigations regarding the pathogenesis of rheumatoid arthritis have been performed using immune cells. However, several studies have demonstrated the involvement of transcription factors expressed in chondrocytes, such as AT-rich interactive domain 5a (Arid5a), Arid5b, and IκBζ, in rheumatoid arthritis. Arid5a was shown to play an important role in stabilization of *IL-6* mRNA, thus regulating IL-6 expression [83]. Additionally, Arid5a interacts with Sox9 to stimulate chondrocyte differentiation [84]. Although the involvement of Arid5a in pathogenesis of rheumatoid arthritis is still unknown, it might be an important transcription factor during onset. Another Arid family transcription factor, Arid5b, has also been proposed as a risk factor for rheumatoid arthritis [85]. Indeed, Arid5b was demonstrated to promote endochondral ossification by controlling epigenetic action in cooperation with Sox9 and PHD finger protein 2, a H3K9 histone demethylase [86]. 

IκBζ, which plays an important role in the differentiation of IL-17-producing helper T cells, is reportedly involved in rheumatoid arthritis through interactions with ROR nuclear receptors [87]. Although the precise relationship between NF-κB and IκBζ remains elusive, (both of which are activated by IL-1 [31]), the latter may be a potential therapeutic target for rheumatoid arthritis. 

## 9. Signal Transduction in Fibrodysplasia Ossificans Progressive (FOP)

Progressive heterotopic ossification is pathogenic, while ectopic bone formation in muscle and tendon mostly occur after exogenous stimuli, such as injury [88]. The process of progressive heterotopic ossification is likely to be excessive endochondral ossification [89]. FOP is an autosomal dominant disorder and one of the major progressive forms of heterotopic ossification [89]. Most progressive heterotopic ossification is caused by excessive BMP signaling [90], while gain-of-function mutations of activin A receptor type 1 (*Acvr1*, also known as *ALK2*), which mediates BMP signaling, are responsible for FOP (Figure 2) [91]. Indeed, several mutations in the *ACVR1* gene have been identified in patients with FOP [92]. As Acvt1 belongs to the BMP type I receptor family, these mutations mediate cytoplasmic signals even in the absence of ligands [93]. A biochemical study showed that a gain-of-function mutation of ACVR1, which causes FOP, activates Smad1 and Smad5 [94]. Moreover, it has been shown that the R206H (arginine 206 mutated to histidine) mutant of ACVR1 activated chondrocyte differentiation without BMP treatment [95]. Taken together, these findings indicate that gain-of-function mutation of the *ACVR1* gene stimulates Smad signaling and promotes BMP-independent chondrogenesis, consequently progressing heterotopic ossification (Figure 2). 

R206H mutant ACVR1 knock-in (KI) mice, which exhibit a FOP-like phenotype, die at birth [96]. Thus, conditional R206H mutant ACVR1-KI mice, which also demonstrate a FOP-like phenotype, were generated [97]. Surprisingly, activin A, which normally antagonizes BMP-Smad1/5 signaling, was identified as a ligand of R206H mutant ACVR1 (Figure 2), but not wild-type ACVR1 [97,98]. Activin A was found to stimulate phosphorylation of Smad1 and Smad5, and promote chondrogenesis through R206H mutant ACRV1 [97,98]. The FOP-like phenotype observed in R206H AVCR1 conditional-KI mice was blocked by treatment with an anti-activin A antibody [97]. Additionally, chondrogenesis of induced pluripotent stem cells (iPSCs) derived from a patient with FOP was inhibited by a specific activin A inhibitor [98]. Recently, inhibitors of mechanistic target of rapamycin kinase (mTOR) have been identified as potential therapeutic reagents for FOP by high-throughput screening of iPSCs derived from FOP patients (Figure 2) [99]. mTOR inhibitors, such as rapamycin, suppressed chondrocyte differentiation of FOP patient-derived iPSCs in the presence of activin A [99]. Further, rapamycin abrogated progressive heterotopic ossification in conditional transgenic mice overexpressing the R206H ACVR1 mutant [99]. mTOR signaling was upregulated along with chondrocyte differentiation of FOP patient-derived iPSCs. Ectonucleotide pyrophosphatase/phosphodiesterase 2 (Enpp2) was found to be involved in the elevation of mTOR activity (Figure 2) [99]. Thus, the activin-A/ACVR1/Enpp2/mTOR axis is likely plays an important role in the pathogenesis of FOP. Notably, as rapamycin has been used for transplantation medicine, it seems plausible that rapamycin can be repositioned for use in FOP patients. 

Other approaches have been performed to develop therapeutic reagents. A phase 2 clinical trial of a human anti-activin A antibody, REGN2477, is currently underway [100]. Palovarotene, an all-trans retinoic acid receptor γ agonist shown to inhibit chondrocyte differentiation [100], is currently in a phase 3 clinical trial for FOP. These developments could contribute to effective therapy not only for FOP, but also for other progressive heterotopic ossification disorders resulting from ectopic chondrogenesis. 

## 10. FGF Receptor 3 (FGFR3) and Achondroplasia (ACH)

ACH, the most common genetic disorder in skeletal tissues, is characterized by short limb dwarfism [20]. Gain-of-function mutations of the *FGFR3* gene result in ACH (Figure 2), as well as related chondrodysplasias including thanatophoric dysplasia (TD) and severe achondroplasia with developmental delay and acanthosis nigricans (SADDAN) [20]. G380R mutant mice, in which the glycine residue of FGFR3 at 380 is replaced by arginine, exhibit a phenotype resembling ACH patients as a result of impaired chondrocyte proliferation and differentiation [101,102]. ACH mutant FGFR3 is known to activate STAT1 in addition to phospholipase Cγ (PLCγ), MAPK, and phosphatidylinositol 3-kinase (PI3K) signaling pathways [20]. Although PLCγ plays a central role in FGF signaling, PLCγ is not necessary for the proliferative action of FGF [103]. Mutation of FGFR3, which causes TD type II, can reportedly abnormally activate STAT1 and induce the cell cycle inhibitor p21, consequently leading to the arrest of chondrocyte proliferation [104]. In contrast, another study indicated that constitutively active meiosis-specific serine/threonine-protein kinase 1 (MEK1), which is upstream of MAPK, in cartilage resulted in an ACH-like phenotype independent of STAT1 [105]. Therefore, it is possible that both STAT1 and MAPK pathways account for the reduction of chondrocyte proliferation observed in ACH and TD (Figure 3). 

Snail family zinc finger 1 (Snail1), which represses type 2 collagen and aggrecan expression [106], has been shown to mediate FGFR3 signaling [107,108]. Importantly, overexpression of Snail1 in mice caused an ACH-like phenotype, reduced chondrocyte proliferation, and induced nuclear localization of STAT1 and *p21* expression [107]. Snail1 overexpression also activated MAPK and impaired chondrocyte maturation. Snail1 expression was found to be markedly upregulated in TD type II mutant mice [107]. Thus, Snail1 is an important downstream mediator of FGFR3 signaling implicated in the impairment of chondrocyte proliferation and differentiation, presumably through STAT1 and MAPK, respectively (Figure 3) [20,107]. C-type natriuretic peptide (CNP) has also been implicated in endochondral ossification [109]. Interestingly, CNP overexpression in chondrocytes has been demonstrated to rescue ACH-like phenotypes through MAPK [110], however, CNP showed no effect on STAT1 [110]. Unfortunately, defining precise roles of STAT1 and MAPK signaling in the pathogenesis of ACH and related chondrodysplasia remains complicated. 

TD and SADDAN, two severe forms of chondrodysplasia, are mostly lethal [20]. TD type I and type II are caused by mutations in the FGFR3 gene at R248, S249, G370, Y373, and K650. TD mutations cause fully constitutively active forms of FGFR3 in a ligand-independent manner [101]. K650M mutation of FGFR3 is responsible for the onset of SADDAN [111]. These TD and SADDAN mutations lead to strong activation of STAT1 and MAPK [111,112]. Importantly, mouse models of TD and SADDAN have been developed [113,114,115].

Several therapeutic approaches for ACH have been proposed. Although administration of growth hormone (GH) reportedly improved dwarfism phenotypes in ACH [116,117], this treatment appears to be insufficient. FGFR3 inhibitors represent a potential treatment for ACH [20]. However, as the specificity of receptor type tyrosine kinase inhibitors is still not established, further investigations are necessary for these approaches. Interestingly, an in vitro study reported that an ACH-related mutation in FGFR3 resulted in the acceleration of apoptosis in a chondrogenic cell line ATDC5 [118]. Moreover, apoptotic changes induced by FGFR3 mutations were inhibited by PTHrP treatment [118]. Consistent with these findings, administration of an active portion of PTHrP (1–34) rescued the ACH phenotype of a mouse model [119]. However, application of PTHrP for ACH patients is unrealistic because PTH and PTHrP therapy have the risk of osteosarcoma, especially in children and young people [120]. Notably, statins, a class of cholesterol-lowering drugs, were identified as an effective reagent for ACH in a study using iPSCs derived from ACH and TD patients [121]. Moreover, a statin drug rescued ACH-like phenotypes in vivo [121]. Therefore, statins are expected to be repositioned for ACH treatment, but the molecular basis of statins in chondrocyte proliferation and differentiation remains to be addressed. Discovery of CNP as a potential treatment for ACH sounds very convincing, however, the short half-life of CNP remains a critical issue for its application. To solve this issue, a stable analogue of CNP, vosoritide, was developed; a phase 2 trial of vosoritide commenced in 2015. Recently, CNP was reported to be effective for ACH patients with the occurrence of only mild side effects [122]. 

## 11. Conclusions

In recent decades, understanding of the molecular mechanisms underlying cartilage diseases such as osteoarthritis, rheumatoid arthritis, FOP, and ACH has become amazingly advanced. Most notably, molecular biological and genetic studies have revealed pathological roles of specific intracellular signaling pathways and transcription factors in these diseases. Based on these advances, several therapeutic targets have been proposed. However, the most successful approach has been the establishment of antibody therapies for rheumatoid arthritis, which have become causal therapy. Although it was very complicated to develop effective therapies for ACH and FOP several years ago, clinical trials have recently been performed for both diseases. These novel approaches are a brilliant development that might yield promising treatments in the near future. However, identification of effective therapeutic approaches for osteoarthritis is still a long way from achieving the ultimate goal of a cure. As described above, there are many potential therapeutic targets for osteoarthritis. The scope to meniscus, synovia, and subchondral bone is critical to understand the pathogenesis of osteoarthritis [123,124,125,126], and important to establish appropriate therapies for osteoarthritis. Recently, a clinical study demonstrated the presence of hypertrophic chondrocytes and osteophytes in articular cartilage of early stage of patients with knee osteoarthritis [127]. Importantly, this study also showed that these pathological changes caused medial meniscal extrusion and consequent dislocation of the meniscus [127]. Presumably, this process is different from the normal regulation of endochondral ossification, and unidentified molecular mechanisms are involved in the underlying pathological events. Therefore, appropriate investigations based on this clinical observation are necessary to understand the pathological basis of osteoarthritis.

## Figures and Tables

**Figure 1 ijms-21-01340-f001:**
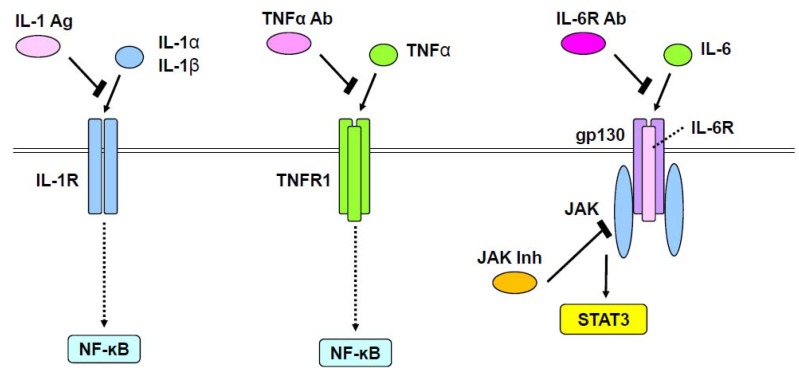
Therapeutic approaches for rheumatoid arthritis. IL-1α and IL-1β, or TNFα stimulate NF-ĸB through IL-1 receptor (IL-1R) or TNF receptor1 (TNFR1), respectively. This action of IL-1α and IL-1β is inhibited by an IL-1 antagonist (IL-1 Ag). IL-6 stimulates the JAK/STAT pathway through a complex of IL-6 receptor (IL-6R) and gp130. Anti-IL-6R antibody (IL-6R Ab) blocks the activity of IL-6. JAK inhibitors (JAK Inh) also block the JAK/STAT pathway.

**Figure 2 ijms-21-01340-f002:**
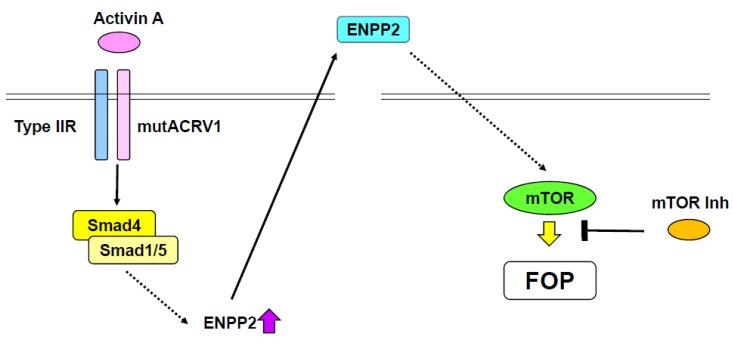
Potential therapy for fibrodysplasia ossificans progressive (FOP). A gain-of-function mutation of ACVR1 (mutACVR1) induces ENPP2 expression by activating the Smad pathway. Activin A functions as a ligand of the mutant ACRV1. Secreted ENPP2 stimulates mTOR signaling, consequently causing FOP. mTOR signaling suppresses the effect of mutant ACRV1.

**Figure 3 ijms-21-01340-f003:**
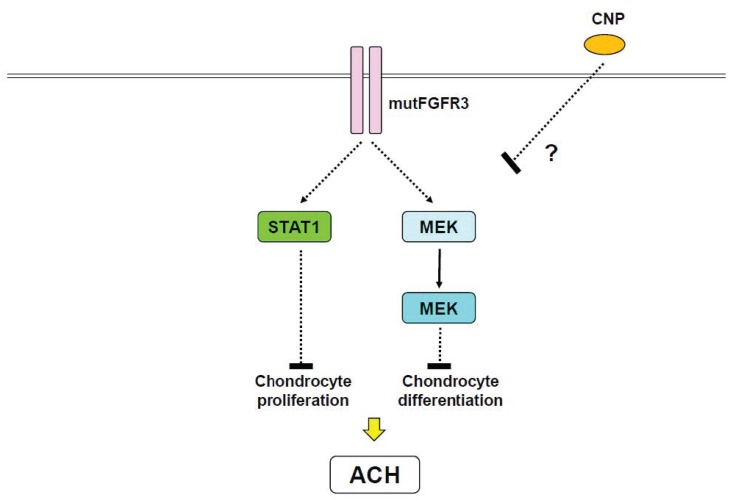
Proposed therapy for achondroplasia (ACH) by C-type natriuretic peptide (CNP). A gain-of-function mutation of FGFR3 (mutFGFR3) stimulates STAT1 and MAPK pathways, both of which inhibit chondrocyte proliferation and differentiation, respectively. CNP inhibits progression of ACH in vivo, however, the precise mechanism is unknown.

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
