# Peer review of "Role of Signal Transduction Pathways and Transcription Factors in Cartilage and Joint Diseases"

_ijms, 2020, doi:10.3390/ijms21041340_

Round 1

Reviewer 1 Report

The revised manuscript is focused on the role of signal transduction pathways and transcription factors in cartilage and joint diseases with a particular focus on Osteoarthritis (OA).

There are still some points that the authors should clarified.

First of all, it is not easy to follow the rationale of the manuscript. Maybe the topic is to wide.

In the introduction, the authors wrote that OA is a mainly caused by excessive mechanical stress, inflammation, and immunological change, is a very common disease of cartilage and joints. I think that they should recognize that OA now is a whole joint disease involving all joint tissues, not only cartilage (also meniscus, synovial membrane, subchondral bone and infrapatellar fat pad). Then, the authors discuss about cytokines in arthritis and OA also as potential therapeutic targets. Afterwards, they wrote that there are several growth factors and cytokines that regulate chondrocyte proliferation and differentiation. Finally they pointed out that there are several mutations causing cartilage related diseases. In my opinion, the aim of the introduction should give a background to the reader to understand and follow the manuscript. Here, I do not understand the connection between the introduction and the rest of the review. For example, the second paragraph is about the role of NF-ĸB in OA but NF-ĸB is not mentioned in the introduction. Second paragraph is about the role of RUNX2 in OA but RUNX2 is not mentioned in the introduction (again). Third paragraph is about a specific miRNA but miRNA are not the focus of the introduction….

The aim of the review is: “In this review, we introduce and discuss our recent understanding of the role of intracellular signaling pathways and transcription factors involved in the molecular pathogenesis of cartilage and joint diseases, as well as potential therapies.” First, it is not clear the part “ molecular pathogenesis of cartilage”. Second, I suggest to clearly define what is the focus of this new review (i.e NF-ĸB in OA, RUNX2 in OA etc).

Inflammatory cytokines are involved (as the authors reported in the introduction) also in OA. However, they discuss only their involvement in RA with a focus as therapeutic targets and there is nothing about cytokines and OA.

Author Response

Reviewer 1

We sincerely thank the reviewer for the critical reading and valuable comments. We believe that these are very helpful to improve our manuscript.

The revised manuscript is focused on the role of signal transduction pathways and transcription factors in cartilage and joint diseases with a particular focus on Osteoarthritis (OA).

There are still some points that the authors should clarified.

First of all, it is not easy to follow the rationale of the manuscript. Maybe the topic is to wide.

The authors agree with the reviewer’s comment that the rationale of the review was not clear, especially in the introduction. We carefully revised the manuscript to make our point clearer. Thank you for your constructive and important comment.

In the introduction, the authors wrote that OA is a mainly caused by excessive mechanical stress, inflammation, and immunological change, is a very common disease of cartilage and joints. I think that they should recognize that OA now is a whole joint disease involving all joint tissues, not only cartilage (also meniscus, synovial membrane, subchondral bone and infrapatellar fat pad). Then, the authors discuss about cytokines in arthritis and OA also as potential therapeutic targets. Afterwards, they wrote that there are several growth factors and cytokines that regulate chondrocyte proliferation and differentiation. Finally they pointed out that there are several mutations causing cartilage related diseases. In my opinion, the aim of the introduction should give a background to the reader to understand and follow the manuscript. Here, I do not understand the connection between the introduction and the rest of the review. For example, the second paragraph is about the role of NF-ĸB in OA but NF-ĸB is not mentioned in the introduction. Second paragraph is about the role of RUNX2 in OA but RUNX2 is not mentioned in the introduction (again). Third paragraph is about a specific miRNA but miRNA are not the focus of the introduction….

These comments are also linked to the previous comment, and we included more information and background in the introduction. We also enhanced the linkage between the introduction and the following parts. We totally agree with the reviewer that “meniscus, synovial membrane, subchondral bone and infrapatellar fat pad” are important issues to understand the pathogenesis of osteoarthritis. However, as we focused on chondrocytes and cartilage in the manuscript, we described this important point in the discussion section. Thank you so much for providing us these important comments.

The aim of the review is: “In this review, we introduce and discuss our recent understanding of the role of intracellular signaling pathways and transcription factors involved in the molecular pathogenesis of cartilage and joint diseases, as well as potential therapies.” First, it is not clear the part “ molecular pathogenesis of cartilage”. Second, I suggest to clearly define what is the focus of this new review (i.e NF-ĸB in OA, RUNX2 in OA etc).

We revised the sentence in the introduction.

Inflammatory cytokines are involved (as the authors reported in the introduction) also in OA. However, they discuss only their involvement in RA with a focus as therapeutic targets and there is nothing about cytokines and OA.

This is also an important point. We described the role of IL-1 in the second paragraph of the original version of the manuscript. However, we agree with the reviewer that it was not clear. To make it clearer, we changed the title of the paragraph and revised the introduction.

Reviewer 2 Report

This review article is worth reading and novel. This article adds depth knowledge about signaling transduction pathways in Cartilage and bone regeneration. However, several issues such as abbreviations, sentence formats (structures), and grammar need to be addressed for further consideration.

For instance, the abbreviations such as FOP, ACH, CNP... need to be expanded. The authors need to refine grammar and sentence structure thoroughly.  

Author Response

Reviewer 2

Comments and Suggestions for Authors

We appreciate the reviewer’s critical and careful reading of the manuscript. We are very pleased that the previously revised manuscript was evaluated as “worth reading and novel”.

This review article is worth reading and novel. This article adds depth knowledge about signaling transduction pathways in Cartilage and bone regeneration. However, several issues such as abbreviations, sentence formats (structures), and grammar need to be addressed for further consideration.

For instance, the abbreviations such as FOP, ACH, CNP... need to be expanded. The authors need to refine grammar and sentence structure thoroughly. 

We carefully checked abbreviations, structure, and grammar in the revised manuscript. The revised manuscript has been also checked by a native commercial editor, again. In the previous manuscript, we described the abbreviations of FOP and ACH in the introduction. However, it would be missing for the readers. To avoid the confusion of the abbreviations, we also described these abbreviations in the heading for each subsection.

Round 2

Reviewer 1 Report

The revision has improved the manuscript. Line 247: there is a typo.

This manuscript is a resubmission of an earlier submission. The following is a list of the peer review reports and author responses from that submission.

Round 1

Reviewer 1 Report

The authors should add a paragraph reporting the literature search method used (databases used, words used including boolean operators, inclusion/exclusion criteria), even if this is a narrative review.

The title should be revised and be more general accordingly to the fact that the review is focused not only on endochondral ossification and cartilage disease.

The authors should add a figure or a table focused on each single transcription factor in order to describe better the function, the genes that are regulated by each factor and the connections.

In some cases it is not clear if the experiments demonstrating the concept were carried out using human tissues or mouse tissues, human or mouse articular chondrocytes (primary or cell lines) in vitro etc. The authors should check and specify which “source” was used if not reported.

The same authors published a recent review (Nishimura R, Hata K, Takahata Y, Murakami T, Nakamura E, Yagi H. Regulation of Cartilage Development and Diseases by Transcription Factors. J Bone Metab. 2017;24(3):147–153. doi:10.11005/jbm.2017.24.3.147) without citing it in this manuscript. The review is quite similar. Therefore, in what way do the authors believe that this new work brings novelty/innovations compared to the old one.

It would be useful to add something new in order to bring novelty and to differentiate the current review from the previous one.

Minor concerns:

The authors should use the same font throughout the manuscript.

Please use italics referring to gene expression.

Reviewer 2 Report

Minor revisions are required to check for spelling, punctuation, consistent use of fonts, reference formatting, and grammar. For example, ‘in vitro’ should be italicized throughout

Page 3, Line 82 consistent font usage…Sox9 are dynamic and exhibit spatiotemporal changes, allowing them to carry out several different…

Page 3, Line 101 spelling: … in these mice [24], it can be deduced that Adamats5 is a direct target of miR-140. Indeed, in vitro…

Reviewer 3 Report

This review imitates the previous articles published by the same authors in different journals. 

1. Riko Nishimura, Kenji Hata, Koichiro Ono, RikakoTakashima, Michiko Yoshida,Toshiyuki Yoneda. Regulation of endochondral ossification by transcription factors. Journal of Oral Biosciences
Volume 54, Issue 4, November 2012, Pages 180-183. https://doi.org/10.1016/j.job.2012.09.001

2. Riko Nishimura, Kenji Hata, Koichiro Ono, Katsuhiko Amano, Yoko Takigawa, Makoto Wakabayashi, Rikako Takashima, Toshiyuki Yoneda. Regulation of endochondral ossification by transcription factors. Frontiers In Bioscience, 17, 2657 - 2666, 2012. DOI No:10.2741/4076

3.Regulation of Cartilage Development and Diseases
by Transcription Factors. J Bone Metab 2017;24:147-153
https://doi.org/10.11005/jbm.2017.24.3.147

4.Regulation of transcriptional network system during bone and cartilage development. Journal of Oral Biosciences
Volume 57, Issue 4, November 2015, Pages 165-170
https://doi.org/10.1016/j.job.2015.06.001 

5.Transcriptional network systems in cartilage development and disease. Histochemistry and Cell Biology.
April 2018, Volume 149, Issue 4, pp 353–363

Therefore, this review article has no novelty since the claimed objectives are already published by this author. Also, this manuscript does not provide a novel insight/impact, hence I advise the authors to put more effort into review article writing-collect advance study reports with a new concept.